# Nutritional composition, fatty acids profile and immunoglobulin G concentrations of mare milk of the Chilean Corralero horse breed

M. Jordana Rivero[1,2] * , Andrew S. Cooke[3], Monica Gandarillas[4], Roberto Leon[2],
Veronica M. Merino[5], Alejandro Velásquez[2,6]

**1** Net Zero & Resilient Farming, Rothamsted Research, Okehampton, United Kingdom, **2** Departamento de Ciencias Agropecuarias y Acuícolas, Facultad de Recursos Naturales, Universidad Católica de Temuco, Temuco, Chile, **3** Department of Life Sciences, School of Natural Sciences, College of Health and Science, Lincoln, United Kingdom, **4** Institute of Animal Production, Faculty of Agricultural and Food Sciences, Universidad Austral de Chile, Valdivia, Chile, **5** Department of Animal Production, Faculty of Agronomy, Universidad de Concepción, Concepción, Chile, **6** Núcleo de Investigación en Producción Alimentaria, Universidad Católica de Temuco, Temuco, Chile

☯ These authors contributed equally to this work.
* jordana.rivero-viera@rothamsted.ac.uk

**Data Availability Statement:** All relevant data are within the manuscript and its Supporting information files.

## Abstract

The objective of the present study was to characterize the nutritional composition, fatty acid profile, and IgG concentration of the milk produced by Chilean Corralero horse (CCH) mares from breeding farms located in southern Chile. Forty-five milk samples were collected from three of the biggest breeding farms (coded as A, B and C) specialized in breeding and selection of CCH in Chile (15 mares sampled per farm). Farms differed in days in milk (DIM). A negative association between DIM and ash, milk protein, milk solids, saturated fatty acids (SFA), and gross energy (GE) was found, whereas DIM had a positive association with monounsaturated fatty acids (MUFA). Milk components like fat, lactose, and energy content varied independently of DIM, indicating other influencing factors such as farm-specific management practices. Offspring sex moderately affected GE content, with milk from mares bearing female offspring having higher GE. Macronutrient profiles of the CCH mares' milk were within the reported range for other horse breeds but tended to have lower fat and total solids. Compared to cow and human milk, horse milk is richer in lactose and lower in fat and protein. Immunoglobulin G concentration was only affected by the farm (B > A) which could be linked to dietary factors and pasture composition rather than maternal parity or other known factors. Overall, CCH mare milk has notable nutritional characteristics, with implications for both foal health and potential human consumption, posing less cardiac risk compared to cow's milk as indicated by lower atherogenic and thrombogenic indices.

**Funding:** Rothamsted Research receives strategic funding from the Biotechnological and Biological Sciences Research Council (BBSRC) of the United Kingdom. Support in writing up the work was greatly received by BBSRC through the strategic program Soil to Nutrition (S2N; BBS/E/C/000I0320) and Growing Health (BB/X010953/1) by MJR at Rothamsted Research.

**Competing interests:** The authors have declared that no competing interests exist.

## Introduction

Milk of mare (*Equus caballus*), and particularly its nutritional composition, has been studied for over 50 years, with the number of published studies steadily increasing, especially in the last decade. A search conducted on the Web of Science website (https://www.webofscience.com, search on 09/05/2023), looking for research articles that contained "mare" OR "horse" AND "milk" in their titles, revealed that 315 studies were published from 1970 to 2022. Within the first two decades, 50 studies were published (i.e., 2.5 per annum), and from 1990 to 2010, 111 studies were published (i.e., 5.3 per annum). From 2011 to 2022, 154 studies were published, equivalent to a rate of 12.8 studies per annum. The recent surge in interest is due to the various nutritional and therapeutic properties associated with mare's milk, which make it beneficial for humans.

In terms of nutritional composition, mare milk has greater protein and lactose concentrations but lower fat concentration compared to cow's milk [1, 2]. Furthermore, mare milk is considered a functional food for humans. Among its properties, mare milk has been found to play a protective role in certain illnesses such as Crohn's Disease, ulcerative colitis, gastric ulcers [3], severe IgE-mediated cow's milk allergy [4], and as a support to the immune system, especially in oncological therapies [5]. Mare and human milks share certain characteristics. Firstly, they are both the natural source of nutrients for foals and babies, respectively, during their first months of development [4]. Milk provides natural sugars that are easily and quickly metabolized, as well as small amounts of fat, predominantly Omega-3 and Omega-6 fatty acids. These fatty acids facilitate the absorption of different vitamins and minerals and help lowering cholesterol levels [1, 6]. Mare milk has a high biological value for human health, as it contains all nine essential amino acids required in the human diet [7]. Mare's milk also contains a higher level of immunoglobulin A (IgA) compared to human and cow's milk. IgA acts as a marker for potentially pathogenic microorganisms, thereby aiding in defense mechanisms [3]. Additionally, mare milk is characterized by the ability to protect and regenerate the skin, particularly in cases of psoriasis, eczema, and atopic skin [1, 8, 9]. Mare's milk also contains proteins that, when digested, release bioactive peptides with blood pressure regulation, antimicrobial, and anti-inflammatory properties [9, 10]. The energy derived from mare's milk primarily comes from lactose [58–70 g/kg] rather than fat [5–20 g/kg] [3]. Therefore, while mare's milk has a lower caloric content compared to cow milk, the majority of these calories come from sugars rather than fats [1, 9, 11]. Additionally, mare milk produces active substances that help regulate the intestinal flora, acting as a prebiotic by limiting the growth of undesirable bacteria and promoting beneficial gut microorganisms [8].

Although all these properties are described for the equine species, milk chemical composition varies based on the mare's age, days in milk (DIM), and season [12]. Furthermore, variations in milk components exist among different breeds [9] such as Andalusian, Arabian [13], Quarter [14], Thoroughbreds [15], Lusitanos [16], and Shetland [17], to name a few. These breeds have been characterized in terms of milk solids, milk protein, milk fat, and lactose concentrations and profiles.

The Chilean horse, designated by supreme decree as a natural monument of Chile according to Prado [18], arrived in America during Cristobal Columbus' second trip in 1493. Since then, it has multiplied exceptionally and later resettled in large numbers in Jamaica and Mexico, eventually spreading throughout the American continent. Prado [18] notes that the diversity that existed in the fifteenth century was later condensed into three distinct and well-defined types, each forming a breed with unique characteristics. The Chilean Corralero horse (CCH), also known as the 'Fina Sangre Chilena' breed in Spanish, primarily originates from the Spanish type. According to Prado [18], its main distinguishing features are: speed,

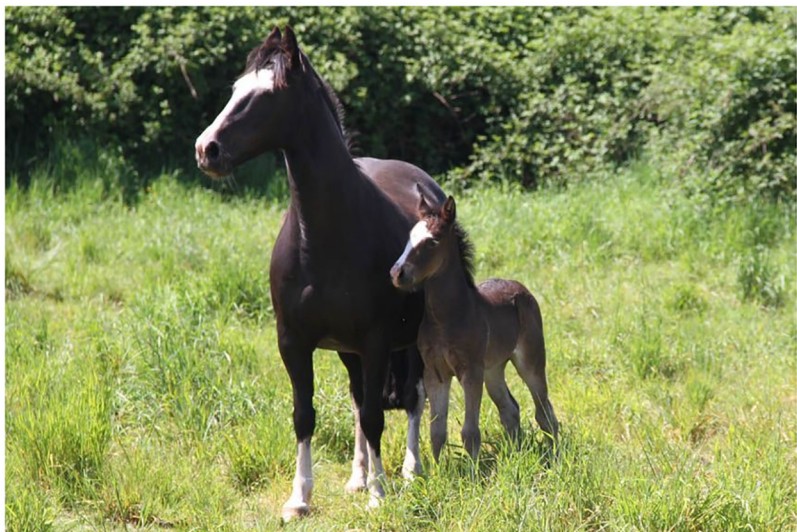

**Fig 1. Mare and foal of the Chilean Corralero horse breed in a breeding farm (photo: Felipe Keim).**

endurance, willingness to take risks, graceful movement, courage, docility, and obedience. The cultural significance of the CCH is immense due to its integral role in Chilean rodeo, which was declared a national sport in 1962 and is recognized by the Olympic Committee of Chile. Rodeo is based on the practical, real-life work that cowboys have to cull a steer from the herd and encompasses a unique set of customs. After football, rodeo is the second most popular sport in Chile [19].

Despite the importance of this breed in Chile, to our knowledge, no published studies have investigated the nutritional or immunological properties of CCH mare's milk. Given the potential benefits associated with mare's milk, this presents an opportunity for innovation and the development of new functional foods. Consequently, the objective of this study was to characterize the nutritional composition, fatty acid profile, and IgG concentration of the milk produced by Chilean Corralero horse mares (Fig 1) from breeding farms located in southern Chile.

## Materials and methods

All the procedures, including animal care and handling procedures, followed national legislation (Law No. 20,380 on Protection of Animals; Decree No. 29 about regulation on the protection of animals during their industrial production, their commercialization and in other areas to hold animals), whose application is supervised by the National Service of Agriculture and Livestock (SAG), the competent authority in this matter. Additionally, this study was carried out in strict accordance with the guidelines of the Committee for the Ethical Use of Animals in Experiments of the Universidad Católica de Temuco. This investigation assessed the nutritional and immunological properties of milk samples from three of the biggest breeding farms (coded as A, B and C) specialized in breeding and selection of CCH in Chile. These breeding farms were located in Los Ríos Region, Chile, one of the main regions breeding these horses. Fifteen samples were collected from each farm, totaling 45 mares sampled (15 per farm) (Fig 2). All the mares were registered in the studbooks of the National Agriculture Society. The mares selected were aged between 4 and 13 years, preferably in their first to fifth foaling, with approximate live weight between 400 and 450 kg. They were all healthy and in good clinical

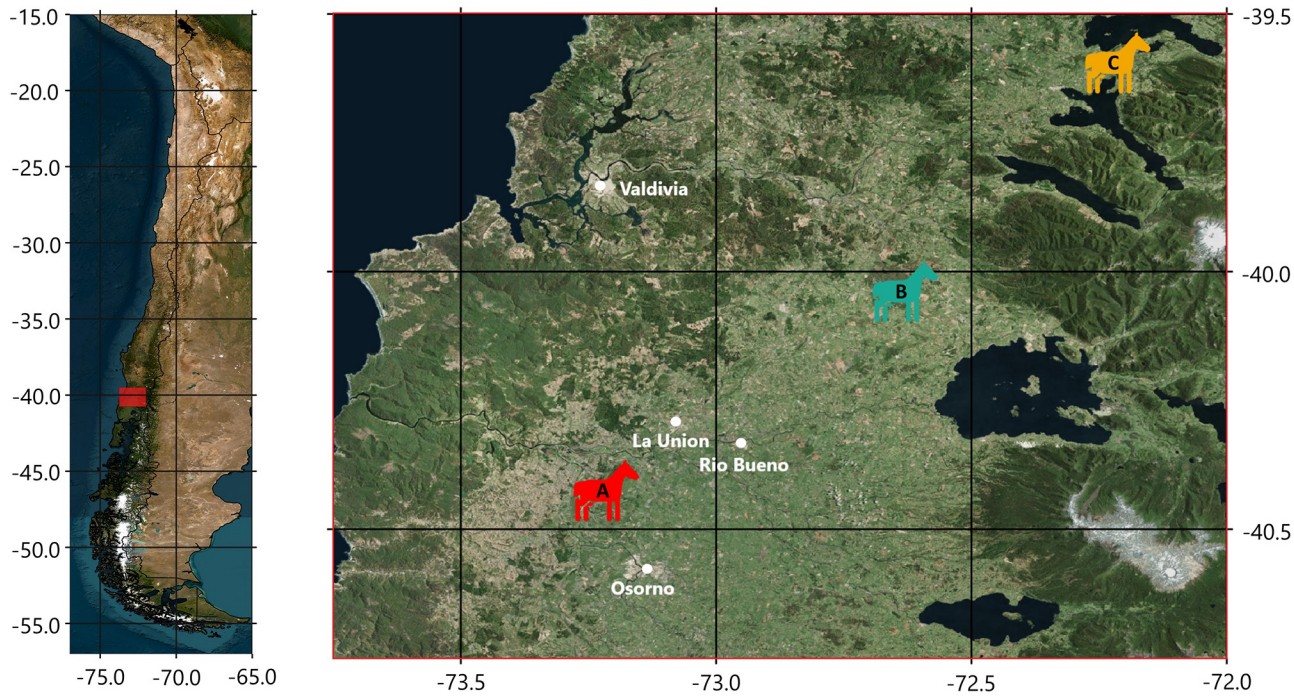

**Fig 2. Location of the three breeding farms: A (red icon), B (green icon) and C (orange icon).** Map image is the intellectual property of Esri and is used herein under license. Copyright © 2020 Esri and its licensors. All rights reserved.

and reproductive conditions. A set of descriptive statistics to characterize the mare's population is shown in Fig 3.

## Description and mare management of the breeding farms

Mare's milk samples for this study were collected from three commercial CCH breeding farms, identified as A, B and C farms (approximate location shown in Fig 2). The sampling took place within a window of time of 13 days (i.e., between the first and the third breeding farm visited) in November 2016 (late spring).

The farm *"A"* was located in San Pablo town which belongs to Los Lagos Region (Central Valley), where temperatures are warmer in winter and summer; the mean annual temperature is 11.9˚C, with a mean of 17.3˚C in January (summer in the south hemisphere, the hottest month of the year) and 7.5˚C in July (winter, the coldest month). The climatic average annual rainfall (1986–2015) for the area was 1195 mm pa while the rainfall accumulated from January to November 2016 was 659 mm (source: NASA POWER Data Access Viewer (nasa.gov) drawn from MERRA-2). The mares are kept under a more intensive system than in farm B and C. They are fed in paddocks of less than 4 ha, with rotation every 4 days. Each paddock contained shelters for protection against frost, rain or high temperatures, ensuring greater comfort for the animals during their rotation.

The farm *"B"* was located near Futrono town which belong to Los Ríos Region (Andes Mountain foothills), a temperate rainy climate where temperatures are cool in both winter and summer; the mean annual temperature in Futrono is 11.4˚C, with a mean of 15.8˚C in January and 7.4˚C in July. The climatic average annual rainfall (1986–2015) for the area was 1105 mm pa while the rainfall accumulated from January to November 2016 was 770 mm. The mares are managed under an extensive raising system, fed exclusively on permanent pasturelands with

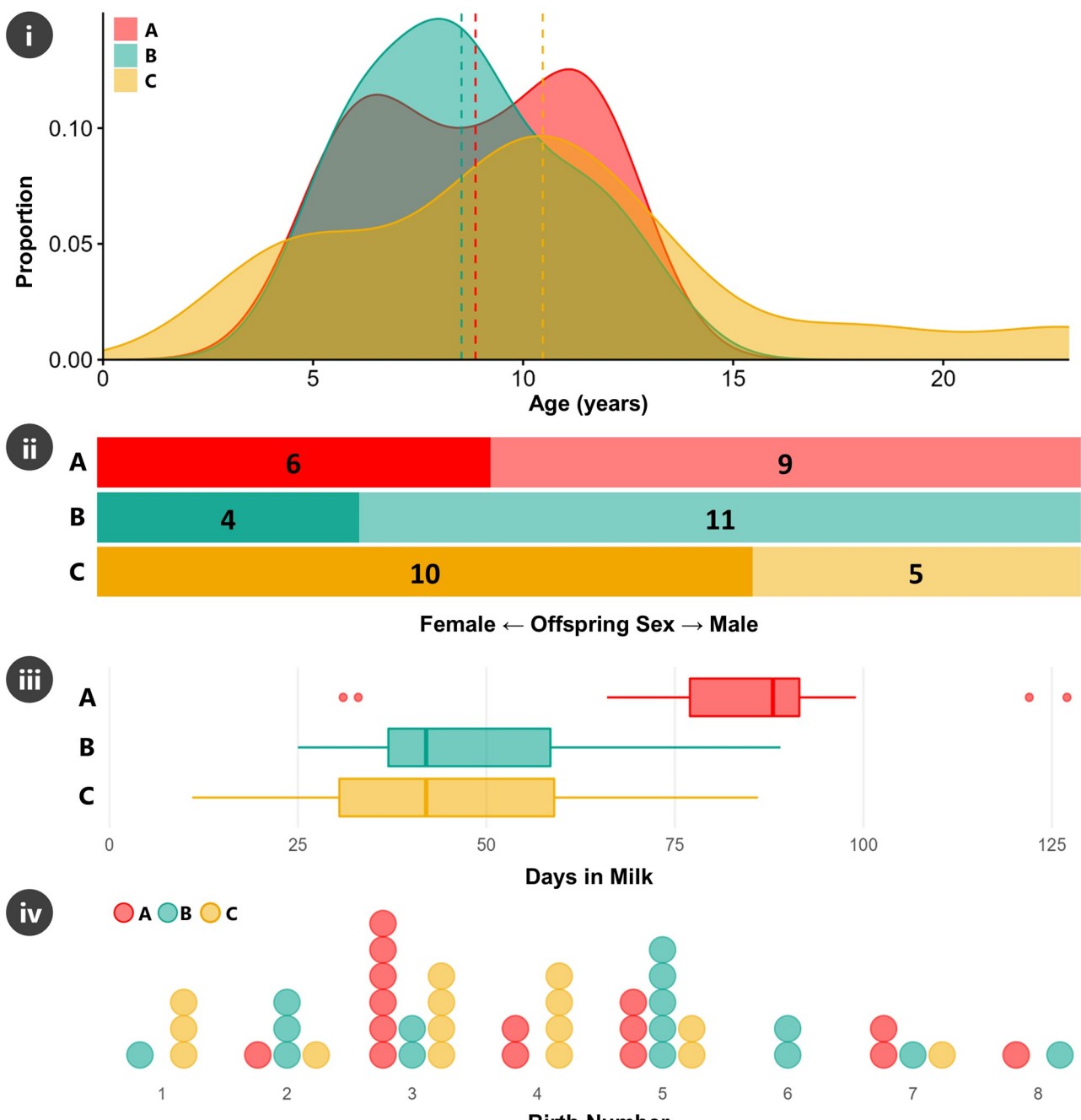

**Fig 3.** Infographic summarizing the three study breeding farms (A, B, C). A) Histogram of age of mares in years. Dashed vertical line is the mean. B) Proportion of sex of offspring for each farm, values are total/count. Left side (darker) are female offspring and right side are male. C) Boxplot showing distribution of the number of the days in milk (DIM). D) Dotplot of the birth number (e.g., total number of offspring the mare has had) for each farm, with each dot representing a single mare.

mineral salt blocks (Block Horse, Quality Pro, Chile) available in feeders in each paddock. They are rotated among paddocks larger than 20 hectares every 20 to 30 days. The paddocks were not equipped with shelters or roofs for protection against frost, rain or high temperatures.

The farm *"C"* was located in Los Ríos Region (10 km from Panguipulli town), with a warm climate and high annual rainfall, which continues even during the drier month. The mean annual temperature is 11.2˚C; in January the mean air temperature is 16.3˚C and in July 6.8˚C. The climatic average annual rainfall (1986–2015) for the area was 1156 mm pa while the rainfall accumulated from January to November 2016 was 786 mm. This farm is near to Panguipulli Lake, which acts as a natural temperature regulator. The mares are fed exclusively on pasturelands, with rotation every 15 to 20 days, and with mineral salt blocks (Equisal®, Schmeisser y CIA Ltda, Chile) available in each paddock. The paddocks were not equipped with shelters or roofs for protection against frost, rain or high temperatures.

## Collection of milk samples

Samples were aseptically collected from the mares, at a volume of 250 mL per mare between 9:30 am and 12:30 pm. Sampling started with the transfer of the mares to pens away from the breeding area (veterinary facilities). They were then separated from their foals which were kept in adjacent pens. The mares were allowed to rest without suckling the foals for 60 to 90 minutes, to allow their udders (alveolar tissue) to fill. Each mare was then taken out of the pen, accompanied by her foal to prevent stress in either animal which could affect the release of milk when the sample was collected. The mare was introduced into a milking stall and the foal was led into an adjacent pen so that the animals remained calm. The mare was milked manually, discarding the first spurts of milk, with all precautions to ensure aseptic conditions: use of gloves, cleaning of the udder, use of new sterilized flasks for the samples. Right after collection, the samples were kept in cool container with icepack to then being stored in the freezer and kept at -18˚C before transferred to the laboratories for analysis.

## Milk components determination

The total solids (TS) content in liquid milk was determined according to the methodology proposed by AOAC [20]. For this, 10 mL of milk were sampled with a volumetric pipette (10 ± 0.02 mL) and placed in a porcelain crucible, then placed in a drying oven [Memmert model UM-500] at 103 ± 2 ˚C for 8 to 24 h, until constant weight. Then, the sample was weighed in analytical balance (Shimadzu AUX-120), and through a mass balance, TS was determined. The crude protein (CP) concentration was determined through the Kjeldahl method [21] with N × 6.38 [22]. Milk fat concentration was determined by the Gerber method [23], using a calibrated butyrometer (Boeco, V1070, Hamburg, Germany). The total ash content (TA) was determined in a mufla oven (62700 Furnace, Thermolyne) by calcination at 550˚C for 8 h [20]. The lactose content was calculated by difference, subtracting the contents of CP, milk fat and ash from the total solids. Subsample from each mare milk sample were freeze-dried (Alpha 1–4 LD, Christ, Germany). Then, pellets were produced for the determination of the gross energy content (GE) in a calorimetric pump (R.S. JESSUP) using the ASTM D2015-66 method [24].

## Determination of milk fatty acids concentrations

Fatty acids were methylated following the method proposed by Morrison and Smith [25] and were separated using a Hewlett Packard 589 series II Plus, (Wilmington, NC, USA) gas chromatograph with a capillary column of 30 m x 0.25 mm x 0.20 μm (SPTM 2380, SUPELCO, Bellafonte, PA, USA). Helium gas was used as a transporting agent. The fatty acids were identified by comparing them with a SUPELCO 37 fatty acid reference standard (Sigma Aldrich, St. Louis, MO, USA). Fatty acids were expressed as a percentage of all identified fatty acids (wet basis). Individual fatty acids were used for computation of the main classes, such as

saturated (SFA), monounsaturated (MUFA) and polyunsaturated (PUFA) fatty acids. The atherogenicity index (AI) and thrombogenicity index (TI) [26] were calculated as below. The n-3 fatty acids included in the calculations were C18:3n3, C20:3n3, C20:5n3, C22:6n3 and the n-6 fatty acids included were C18:2n6c, C18:3n6, C20:3n6, C20:4n6.

$$AI = \frac{C12:0 + (4 \times C14:0) + C16:0}{\Sigma\mathrm{MUFA} + \Sigma\mathrm{MUFA}} \quad (1)$$

$$TI = \frac{C14:0 + C16:0 + C18:0}{(0.5 \times \Sigma\mathrm{MUFA}) + (3 \times \Sigma n - 3) + (0.5 \times \Sigma n - 6) + (\Sigma n - 3 \div \Sigma n - 6)} \quad (2)$$

## Determination of milk IgG concentration

Samples of 50 mL were thawed and homogenized at room temperature after which 100 μl was pipetted into a tube to which 900 μl of sample dilution buffer was added (1:10 diluted sample). After homogenizing on a vortex for 2 sec, 100 μl of the 1:10 diluted sample was pipetted into a tube and 900 μl sample dilution buffer was added and homogenized on a vortex for 2 sec (1:100 diluted sample). Another 10 times dilution was applied to obtain a 1:1000 diluted sample [27]. The samples were then analyzed using the ELISA test (Double Antibody Sandwich) [28] to determine the IgG concentration.

## Samples collection and chemical analysis of forage

To characterize the nutritional quality of the forage on which all the mares sampled were fed, chemical analysis was carried out on samples of the different pasturelands (Table 1). In each breeding farm, twelve herbage sub-samples were cut manually at a height of 5 cm following a zig-zag pattern within the paddock where the mares were grazing at the sampling time. The 12 sub-samples pooled to obtain a homogenized sample of 1 kg of fresh material and were stored in hermetically sealed Zeplock® bags, eliminating as much air as possible. They were then kept in cold chambers at -18˚C and transported to the laboratory. The dry matter (DM) was determined by drying in an oven with forced air at 60˚C (Binder, FD 53 #05–76079) for 48–72 h to constant weight [20]. The CP (N x 6.25) concentration was determined by the methodology of Kjeldahl [20]. The total ash concentration was determined by calcination in a muffle furnace (62700 Furnace, Thermolyne) at 550˚C for 8 h [20]. The neutral detergent fibre (NDF) and acid detergent fibre (ADF) concentrations were determined by the modified ANKOM method [29]. Fibrous fractions were determined with alpha amylase, expressed exclusive of residual ash.

**Table 1. Nutritional composition (dry matter (DM) basis) of the naturalized pasture where the mares grazed for each breeding farm (A, B and C).**

| | Breeding farms identifiers | | |
|---|---|---|---|
| Nutritional Variable | A | B | C |
| Dry matter content (g/100 g) | 21.1 | 19.6 | 23.4 |
| Total ash (% DM) | 8.52 | 8.89 | 8.47 |
| Crude protein (% DM) | 17.9 | 21.0 | 15.0 |
| Neutral Detergent Fibre (% DM) | 52.0 | 50.5 | 52.6 |
| Acid Detergent Fibre (% DM) | 25.5 | 24.5 | 27.2 |

**Table 2. Overall means of mare milk nutritional components and comparison across factors.**

|  | Ash | Fat | Lactose | Protein | Solids | Non-fat solids | IgG | GE |
|---|---|---|---|---|---|---|---|---|
| **Mean$^{(\pm C.I.)}$** | 0.42$^{(0.02)}$ | 0.99$^{(0.16)}$ | 6.4$^{(0.11)}$ | 2.08$^{(0.09)}$ | 9.89$^{(0.16)}$ | 8.91$^{(0.15)}$ | 1159$^{(98.7)}$ | 1787$^{(48.7)}$ |
| **Farm (B-A)** | -1.41 | -4.37*** | 3.12** | -0.71 | -2.26* | 2.20* | 4.06*** | -3.30** |
| **Farm (C-A)** | 0.38 | -3.53*** | 3.27** | 1.30 | -0.20 | 3.70*** | 0.96 | -2.68* |
| **Age** | -0.15 | 1.76^ | -1.57 | 1.08 | 0.97 | -0.82 | 0.01 | 1.05 |
| **Birth n.** | -0.28 | -0.60 | 0.54 | -0.98 | -0.66 | -0.11 | 0.41 | 0.58 |
| **Offspring sex (M-F)** | -1.02 | -0.60 | -0.67 | 0.14 | -1.12 | -0.65 | -0.88 | -2.12* |
| **DIM** | -6.54*** | -1.02 | 1.94 | -3.98*** | -2.12* | -1.36 | 0.23 | -2.24* |

Top: Mean values for macronutrient (g/100 g otherwise stated), solids, non-fat solids, immunoglobulin G (IgG, mg/L), and gross energy contents (GE, J/g) of milk, superscript values represent 95% confidence interval. Bottom: GLM results (*t*-values) comparing milk nutritional components across milk/mare factors. Colors correlate with *t*-value. Superscript values are *p*-values: 0 '***' 0.001 '**' 0.01 '*' 0.05 '.' 0.1 ' ^'. DIM: days in milk.

## Data analysis

A Chi-square analysis was performed to assess whether the farms differed in the characteristics of the mares sampled with regard to birth number (1–3 vs 4+), age (4–9 vs 10+) or sex of the foal (male or female). An unbalanced ANOVA was applied to the DIM and age variables in response to combinations of farm and sex of the offspring. Principal component analysis (PCA) was conducted using the macronutrient (ash, fat, lactose, protein) results of each milk sample. To enable comparison with other milk sources, macronutrient data from other mares and other species, taken from the literature, were also plotted within this PCA. General linear models (GLM) were conducted to assess differences in macronutrients, solids, non-fat solids, IgG, individual fatty acids and their sums across the variables of farm, age of the mare, total number of off-spring to date, and DIM. Pearson's correlation analysis was performed considering all the milk components against one another. Statistical analysis and figures were performed in R (i386 4.1.2) and R Studio (2021.09.1) using packages ggplot2, ggfortify and Cairo.

## Results

### Characteristic of the mares sampled

Simple chi-squared analyses of associations between farm (herd) and birth number, and sex of offspring, showed no significant associations (p > 0.05), though farm C has a bias towards females and the other two farms (particularly B) towards males, with a similar difference in bias between C and A/B for age (Fig 3, Table 2). Significant effects (p < .001) of farm on DIM (82.7 d for A greater than 47.8 d for B and 43.7 d for C, Fig 3) was found, but not on age (p > 0.05), averaging 9.29 y.

### Macronutrients and IgG concentrations

There were significant differences among farms in milk composition (Table 2); this was most notable for fat content which was greatest on farm A (1.39 g/100 g, s.d = 0.60) compared to B (0.64 g/100 g, s.d = 0.28) and C (0.94 g/100 g, s.d = 0.40). A strong negative association was observed between DIM and both ash, protein, solids, SFA, and GE, whilst there was a positive association with MUFA (S1 Table). Offspring sex appeared to have a moderate impact on milk energy content; milk from mare's bearing female offspring had greater GE (1858 J/g, s. d = 166.4) compared to those bearing male offspring (1734 J/g, s.d = 115.2). Age and birth number appeared to have no impact on milk composition. IgG averaged 1159 mg/L an the only significant effect observed was due to the farm (B > A, Table 2).

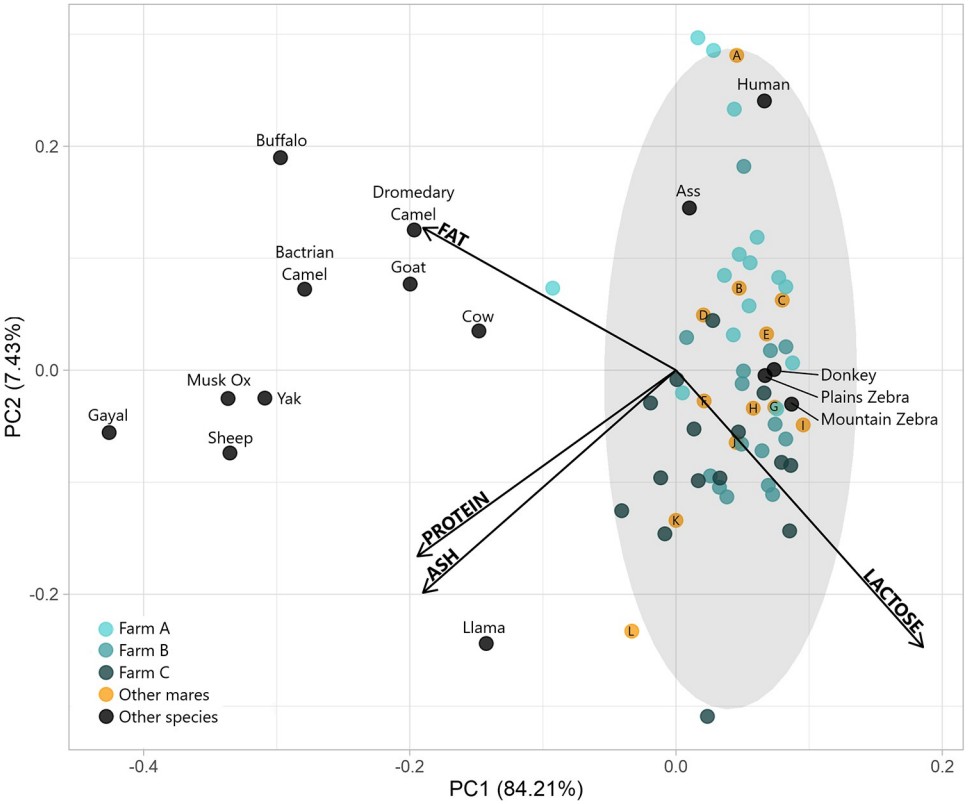

**Fig 4.** Principal Component Analysis (PCA) of macronutrient (ash, fat, lactose protein) profiles of Chilean Corralero horse milk (collected from the breeding farms A, B and C) alongside examples of milk from other mares and species taken from the literature. The gray oval is a 95% confidence ellipse for Chilean Corralero horse milk. Yellow points with letters represent examples of mare's milk from the literature, breeds as follows: A: Halfinger [30], B: Mongolian [31], C: Przewalski [32], D: Murgese [33], E: American Quarter [30], F: Italian Saddle [34], G: Halfinger [35], H: Konik [36], J: Pony [32], K: Polish Coldblood [37], L: Unspecified [1]. Black points represent milk from other species: Ass [32], Bactrian Camel [38], Buffalo [38], Cow [38], Donkey [38], Dromedary Camel [38], Gayal [38], Goat [39], Human [39], Llama [38], Mountain Zebra [32], Musk Ox [38], Plains Zebra [32], Sheep [39], Yak [38].

Only a few strong correlations coefficients were found to be significant (S1 Table); milk fat and lactose were negatively correlated (r = 0.710), whereas milk protein was directly associated with total solids (r = 0.717) and ash (r = 0.919). Similarly, GE was positively correlated with milk fat (r = 0.753) and total solids (0.814).

Macronutrient profiles (fat, protein, lactose and ash) appeared comparable to those reported in the literature for other mares, and other equids (Fig 4). Equid milks had a distinct profile compared to other milks; PC1 accounted for 84.2% of variation and on that axis; there was no overlap of equine milks (e.g., horse, ass, donkey) with other milks, with the exception of human milk. Compared to other species' milks, equine milks had a lower fat, protein, and ash content, but greater lactose content (Fig 4).

## Fatty acids profile

The main factor affecting fatty acid concentrations was farm (Table 3). Whilst there appeared to be little overall difference in SFA on the whole, there were differences with individual SFA, for example C10:0 concentrations were lower on farm B and C, compared to A (S1 Table). Farm C had lower MUFA concentrations but higher PUFA (S1 Dataset). DIM also had strong

**Table 3. Overall means of fatty acids and fatty acids groups and comparison across factors.**

| | Sum of fatty acids | | | Main SFA | | | | | Main MUFA | | | Main PUFA | | | Health indices | |
|---|---|---|---|---|---|---|---|---|---|---|---|---|---|---|---|---|
| | SFA | MUFA | PUFA | C10:0 | C12:0 | C14:0 | C16:0 | C18:0 | C14:1 | C16:1 | C18:1N9 | C18:2N6 | C18:3N3 | C20:3N6 | AI | TI |
| Mean | 40.3 | 28.9 | 30.8 | 2.92 | 6.11 | 6.97 | 22.1 | 1.72 | 0.64 | 6.37 | 21.7 | 5.51 | 24.3 | 0.72 | 0.95 | 0.45 |
| Median | 4.03 | 28.3 | 31.1 | 3.04 | 6.05 | 7.03 | 21.9 | 1.58 | 0.65 | 6.17 | 21.2 | 5.30 | 24.5 | 0.77 | 0.94 | 0.42 |
| ±C.I. | 1.34 | 1.52 | 1.22 | 0.37 | 0.45 | 0.37 | 0.59 | 0.14 | 0.05 | 0.45 | 1.23 | 0.28 | 1.04 | 1.11 | 0.06 | 0.03 |
| Farm (B-A) | -2.02^ | 0.30 | 2.47* | -5.06*** | -2.66* | -1.28 | 0.87 | 1.45 | 0.77 | -0.39 | 0.38 | 2.57* | 1.93^ | 1.70^ | -1.62 | -1.61 |
| Farm (C-A) | -1.11 | -3.09** | 5.55*** | -2.24* | -1.35 | -1.49 | -0.07 | 3.04** | -0.79 | -3.43** | -2.34* | 7.65**** | 4.44*** | 0.04 | -1.41 | -3.83** |
| Age | -0.92 | -0.18 | 1.53 | -0.75 | -0.56 | -0.16 | -0.56 | -2.96** | 1.15 | 0.52 | -0.43 | 0.09 | 1.24 | 1.20 | -0.52 | -1.03 |
| Birth n. | 0.76 | -1.05 | 0.29 | 1.35 | 1.00 | 0.18 | 0.52 | 2.19* | -2.17* | -2.19* | -0.15 | 0.56 | 0.33 | 0.34 | 0.54 | -0.07 |
| Offspring sex (M-F) | 0.92 | -1.56 | 0.71 | 0.47 | 0.08 | 0.52 | 0.64 | -0.06 | 0.34 | -0.68 | -1.79^ | -1.29 | 0.48 | 0.95 | 0.68 | -0.49 |
| DIM | -3.60*** | 2.96** | 1.29 | -1.80 | -3.73*** | -3.59*** | -2.18* | -0.61 | 0.31 | 0.80 | 3.07** | -1.26 | 0.93 | -1.90 | -3.90*** | -2.29* |

Top: Mean values for fatty acids, superscript values represent 95% confidence interval. Bottom: GLM results (*t*-values) comparing fatty acid components across milk/mare factors [farm, age, birth number, offspring sex, days in milk (DIM)]. Colors correlate with *t*-value. Superscript values are *p*-values: 0 '***' 0.001 '**' 0.01 '*' 0.05 '^' 0.1 ' ' 1.

effect on fatty acid concentrations. Milk from mares with more DIM yielded lower concentrations of SFA but higher concentrations of MUFAs. This was characterized by significant reductions in C12:0, C14:0 and C16:0, and a significant increase in C18:1n9. Interestingly, birth number affected some individual fatty acids but not their sum. Regarding fatty acids indices, AI averaged 0/95 and TI averaged 0.35. Both indices were negatively affected by DIM while the latter was greater in farm C compared with farm A (Table 3).

Only two strong correlations were found between individual fatty acids, with a positive association between the SFA C12:0 and C:14 (r = 0.851) and between the PUFA C18:2n6 and C18:3n3 (r = 0.709) (Fig 5). Medium and positive correlations were found between C10:0 and C12:0 (r = 0.650), C14:0 and C:16:0 (r = 0.643), and C16:1 and C18:1n9 (r = 0.588). Regarding inverse associations, C18:1n9 presented medium negative correlation with C14:0 (r = -0.683) and C12:0 (r = -0.663) (Fig 5).

Medium negative associations (-0.600 < r < -0.500) were found between milk fat and C18:0, both milk protein and ash with both C18:1n9 and sum of MUFA, between IgG and C10:0, and between GE and C18:0 (S1 Table). Also, a medium but positive correlation was found between C10:0 and milk fat.

## Discussion

This research aimed at characterizing the CCH breed's milk in terms of its nutritional composition, including fatty acids profile and IgG content. For this purpose, we collected mare milk samples from 45 mares across three breeding farms located in the main CCH breeding region of Southern Chile. With regard to the sample size reported in similar studies, we performed a search on Web of Science (https://www.webofscience.com/ searched on 10/05/2023) including 'milk' AND 'mare*' OR 'horse' in the title AND 'composition' OR 'nutriti*' in the topic, and obtained 129 records. Among these studies, we were able to locate 59 which did study mare's milk composition and reported the number of mares sampled, i.e., samples size (S2 Table). Most of the studies (68%) sampled less than 20 mares in total, moreover, 29% of the studies took samples from 9 mares or less, and only one study was found to have sampled from 45 or more mares from only one farm. Moreover, only 9 studies declared that milk samples were collected from mares with more than one origin (farms, region), and only 5 studies declared 3 or more origins of the mares (only two studies samples from more than 3 origins), whereas 68% declared samples taken from one farm only, while 10 did not inform this. Therefore, our study would be in the top 5% when ranked according to number of mares sampled and in the top 10% when ranked according to the number of sites (studs or farms) sampled, highlighting the relevance of the dataset collated in terms of representativeness of the CCH breed. On the other hand, considering that the composition of mare's milk, particularly protein, lactose and fat content, is affected by lactation stage, i.e., DIM [3, 40], the fact that the farms differed on average DIM by up to 37 at the time of the sampling could have represented both, a disadvantage, since milk traits between farms might not be comparable due to the variability in DIM, and an advantage, i.e., a wider range of DIM was covered among the breeding farms. Furthermore, we sampled from mares of a wide range of ages and foaling number, enhancing the representativeness of the samples collected with the regard to the factors that may influence milk composition.

### Differences in DIM between farms

In addition to DIM, differences found in composition of mare's milk between farms could be attributed to a combination of factors including weather, age, level of milk production, and dietary management differences [40–42]. With the regard to the differences in DIM, breeding

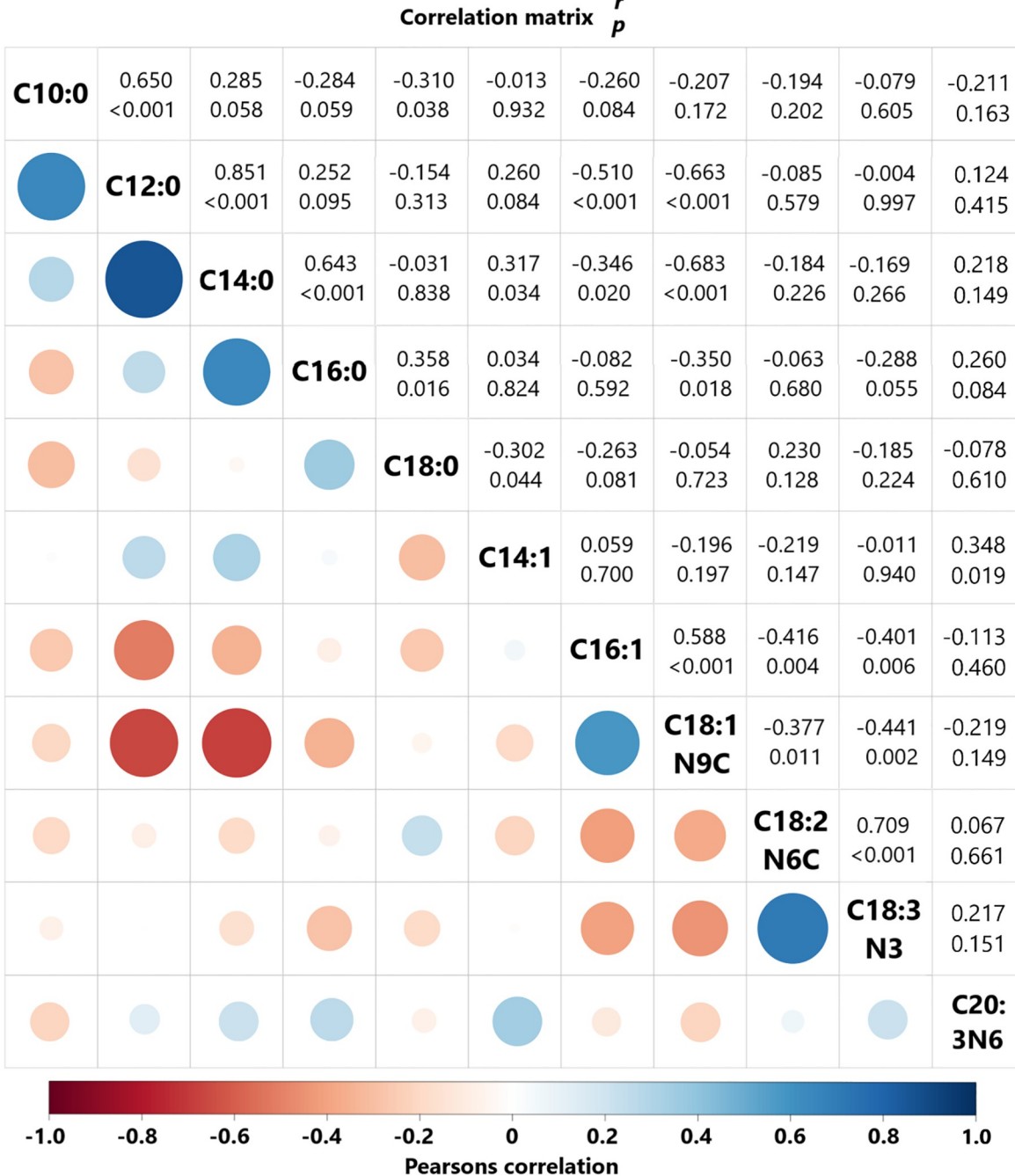

**Fig 5. Pearson's correlation of fatty acid concentrations.** Circle size represents strength of correlation and color represents correlation *r*-value (-1.0 to 1.0). Values in white squares are *r*-values with *p*-values below.

farms could have differed in their reproductive management or in the duration of the gestation [43]. Horses typically have a large variation in their gestation length, usually ranging from 300 to 380 days [44, 45]. It is mainly influenced by environmental and genetic factors including climate, photoperiod, breed, mare age, parity, sex of offspring, season [45–48] and feeding management [49, 50]. For instance, male pregnancies are known to last an average of 2–3 days

longer than female pregnancies [47, 51]. However, the differences observed in the proportion of male and female foals between farms, and in turn their potential effect of length of the gestation, would not explain the differences observed in DIM. Gestation length is also affected by mare age through the variation of the nutritional efficiency of placental function and fetal growth [52]; however, differences in DIM between farms cannot be attributable to mare age since they all had a similar age distribution. Regarding to feeding management, pasture chemical composition in all breeding farms had lower nutritive value (i.e., high levels of NDF) compared with previously reported values for high-quality pastures for dairy cows in southern Chile in spring [53–55]. Although herbage samples were collected from the paddock being grazed by the time when the milk samples were collected, it is important to note that the chemical composition of the farms' pasture arise from one single sampling time, with no physical replicates, therefore differences discussed here are only numerical, as opposed to statistical. Pasture of C farm had higher DM and ADF and lower concentrations of CP than those for A and B, indicating a lower nutritional quality, which may be related to an advanced maturity stage. Energy-restricted diets have been associated with lower reproductive efficiency, increasing the interval to first ovulation and the gestation length than mares fed adequate energy supply [56]. Despite the lower nutritional quality of the C farm pastures, mares from this farm had similar DIM compared to those from the B farm and lower DIM compared to mares from the A farm. This suggests that pasture quality does not account for the differences in DIM. Therefore, it is likely that reproductive management is the primary factor explaining these differences between farms.

## Factors affecting CCH mare milk components contents

Some studies suggest that the concentration of total milk solids, milk fat and milk protein gradually decrease as lactation progresses, while lactose content increases [57, 58], whereas other studies reveal that the concentration of milk fat is not modified by lactation stage when mares are fed with a constant diet [59, 60]. It is interesting to note that DIM *per se* only showed an effect on ash, milk protein, milk solids and GE concentration, with all these components decreasing as lactation progresses. Although farms B and C differed in DIM with farm A, the milk sample origin (i.e., breeding farm) had an independent effect on milk fat, lactose, non-fat solids, and in particular in GE, suggesting there are other factors than DIM imposing variation among the three examined breeding farms. This effect is remarkable with GE; farm A (i.e., the farm with the highest DIM) had higher GE content compared with B and C farms, despite the overall inverse relationship between DIM and GE. This can be explained by the substantially higher milk fat content from mares in A farm, being between 49 and 117% greater than that of C and B farm, respectively. According to Doreau et al. [22], milk fat has 1.71- and 2.18-fold GE than milk protein and lactose, respectively. The decrease in GE content with an increase in DIM observed in our study has been previously reported for other horse breeds [34, 35].

Interestingly, the only milk component affected by sex of the offspring was GE. The combined effect of a higher female offspring proportion and higher milk fat content of mares of farm C would explain the higher GE values compared with farm B. According to Barłowska et al. [42], the milk from mares of female offspring has higher fat and protein concentrations and lower lactose content that mares with male's offspring; however, only the latter is confirmed in our study. In addition to sex of the offspring, the other 'mare' factor showing an effect on milk components was the age of the mare, whereas foaling number, even though is directly related to the age, did not affect milk nutrients content. Hence, milk fat content tended to be higher as mare's become older. However, previous studies on purebred Quarter Horse [14] and on Mangalarga Marchador mares between 8 to 19 years old [61] found no effect of

age on milk fat. Contrastingly, Czyzak-Runowska et al. [40] reported an effect of mare's age on milk lactose content decreasing in the milk of mares after the seventh foaling.

The lower milk fat content observed in B farm in particular could be due to the lower NDF content than pastures from the other two farms. Mare milk fat content mainly depend on the amount of fatty acid from diet and of fatty acids produced by the mammary synthesis from acetate or butyrate [62], end products of fermentation of fibre in caecum and large intestine, that are absorbed and used as source of energy [63]. However, contrary to dairy cattle, little information exists about the effect of pasture quality on the concentration of these volatile fatty acid and its relationship with milk composition in horses. Horses are adapted to ingesting large amounts of forage with high content of fibre due to its caecum and colon function as a fermentation chamber colonized by microbes [64]. Differences in feeding management between farms during the pre-experimental period (late gestation and early lactation) could have also contributed to differences found in milk composition [65, 66]. The concentration of protein in equine colostrum is very high, >150 g/kg, immediately post-partum (due primarily to the high concentration of immunoglobulins) [9]. It has been reported that there is a fast drop in protein level in mares' colostrum during the first hours postpartum followed by a gradual reduction in the normal milk as lactation progresses [6], decreasing below 20 g/kg after 4 weeks of lactation [9], which concurs with our findings in relation to the effect of DIM on milk protein. The decrease in ash content with an increase in DIM observed in our study has been previously reported by Martuzzi et al. [34] comparing early (ca. 19 DIM) to late (ca. 185 DIM) lactation mares of Italian Saddle and Haflinger horse breeds, and also by Schryver et al. [17] of Thoroughbred mares from 1 to 17 weeks post-partum.

## CCH mare milk composition in relation to other horse breeds and mammal species

Several reviews have compiled information regarding the main nutrients of mare milk and how they vary with cow and human milks. Considering the minimum and maximum values in the ranges reported by the reviews, mare milk contains 93–120 g/kg total solids, 14–35 g/kg protein, 3–42 g/kg fat, 55–72 g/kg lactose, 3–5 g/kg ash, which along with lactose seem to be very stable, and 1715–3096 J/g GE, which, along with fat seem to be very variable [1, 9, 67–69]. On average, CCH mare milk falls within the range of values previously reported, with fat, total solids and GE towards the lower values of the range, and with substantial variation due to the origin of the samples (farm), whereas ash and lactose are similar to the mid values of the ranges and with little variation among farms. In general, equid species such as horses, zebras, and asses (genus Equus, family Equidae), including the domestic horse, produce very dilute milks containing only 100 or 110 g solids per kg milk and 10 or 20 g/kg fat [70]. Compared with human milk, horse milk contains less lactose, as much protein, and much less fat, while compared with other milks for human consumption (cow, ewe, goat, camel), horse milk is richer in lactose, lower in protein, and especially lower in fat [69]. When compared with other breeds, CCH mare milk seems to have less fat and total solids than most of the breeds evaluated by Uniacke-Lowe et al. [9] apart from Haflinger and Shetland Pony, and greater fat levels than that of Dutch Saddle, Lusitano and Sella & Salto. On the other hand, CCH mare milk seems to have higher ash content than most of the breeds analyzed by Uniacke-Lowe et al. [9] and more than all the breeds analyzed by Schryver et al., [71], e.g., Shetland Pony, Percheron, Thoroughbred. Similarly, Santos et al. [61] reported higher milk fat (10–10.7 g/kg) and lactose (68.6–71.0 g/kg) contents for the Mangalarga Marchador mares between 40 to 80 DIM. Similarly, Naert et al. [72], evaluating milk composition of mares of different breeds (New Forest, Friesian, Tinker, Warmblood, Shire, and Haflinger) from 10 farms, reported milk lactose content

ranging from 64.4 to 71.6 g/kg. Regarding GE, except for farm B, which had a lower content than A and C farms, most samples were within the range 1816 to 2023 J/g GE reported by Santos et al. [61] for Mangalarga Marchador mares between 40 to 80 DIM. On the other hand, Mariani et al. [35] reported GE content of 1845 and 1757 J/g for Haflinger mares in 40 and 80 DIM, which are lower values and more similar to the ones reported in this study for CCH mare milk. The average milk protein in mares from C farm were similar to the 2.3% reported by Salamon et al. [73] from 29 mares of four breeds (Hungarian Draughts, Haflingers, Bretons and Boulonnais breeds) grazing high quality pastures during their first 45 DIM.

## Fatty acids profile of CCH mare milk

The composition of fatty acids in mare milk is influenced by the dietary intake of fatty acids since they are absorbed in the small intestine without undergoing the initial biohydrogenation process observed in ruminants. In our study, there was a higher concentration of SFA in mare's milk followed by PUFA and MUFA. According to literature data [3, 74, 75], the content of different fatty acids in the fat of mare's milk varies. Pietrzak-Fiecko et al. [76] reported differences between the content of fatty acids in milk fat from mares of various breeds. The milk of Wielkopolski Horse mares was characterized by the prevalence of the unsaturated fatty acids (UFA, 61.3%), which is similar to our findings, whereas in the samples originating from Konik Polski mares SFAs predominated (55.0%). Lower values (46.2%) were reported for UFAs in the Sokólski Breed with access to pasture [42], particularly the PUFAs were low (18.3%) compared with our study.

According to Doreau and Martin-Rosset [69], horse milk exhibits a low stearic acid (C18:0) content, typically less than 2%; however, it contains substantial levels of palmitoleic acid (C16:1), between 2 and 10%, indicating a high activity of mammary Δ9-desaturase. In our study, we found that stearic and palmitoleic acids were within the range reported in the literature. Horse milk is also characterized by a high content in PUFA: linoleic (C18:2n6) and α-linolenic acid (C18:3n3). The latter is very variable (between 2.2 and 26.2) but can reach 25%, especially when the diet primarily consists of forage [77]. This was the case for α-linolenic acid in our study, where the average value surpassed 24%, confirming the relevance of the forage-based diet on fatty acids composition of horse milk. On the other hand, the content of linoleic acid was in the lower part of the range reported for this fatty acid (between 3.6 and 20.3%) [77]. Usually, dietary linoleic acid is present in relevant concentration in concentrates [69], which was not part of the diet of the mares assessed on this study. In fact, the concentration of linoleic and α-linolenic acids recorded in our study (5.51 and 24.2%, respectively) resembles that of these fatty acids in perennial ryegrass forage, i.e., 5.47 and 26.8 mg/g DM, respectively [78]. Butyric, caproic and caprylic acids are usually found in low concentrations in horse milk [77]; however, none of the samples analyzed in our study contained these fatty acids. Moreover, in the case of capric acid (C:10:0), we found values in the lower part of the range reported for this fatty acid (2.3–17.7%). This lower value could be attributed to the breed effect, since it has been reported that this fatty acid varies with breed [76].

Few data are available on minor FA for horse milk. Odd-chain FA, which are characteristic of bacteria, are present in small amounts (between 0 and 1,1%), denoting the low absorbance of FA of microbial origin in the large intestine [77]. In our study, we only detected the odd-chain fatty acid C15:0 (pentadecanoic acid) in one farm, with 11 out of 15 samples containing this FA and averaging 0.30% for that farm (C). The forage fed in this farm had the highest fibre and DM content, and the lowest PC content, indicating a greater level of maturity of the forage. Considering that diet composition is able to influence the gut environment and its functioning [79], it could be assumed that the particular large intestine environment promoted by

the diet may have enhanced the production and absorption of C:15:0. Another minor FA found in milk samples from two farms (farm C 8 samples, average for the farm = 0.18%; farm B 7 samples, average for the farm = 0.23%) was eicosenoic acid (C20:1n9 Cis-11). Partly hydrogenated 20-carbon FA, especially 20:3, 20:2 and 20:1, are present in the range of 0.3 to 0.5% each [77]. Eicosenoic acid is commonly found in various plant oils, e.g., Chia (*Salvia hispanica*), Camelina (*Cameline sativa*) [80] and it has been also found in forage assessed grazing systems based in rye (*Secale cereale* L.) and wheat (*Triticum aestivum* L.). This fatty acid could have been present in plant species comprising the extensively managed pastureland grazed by the mares.

There were strong farm-level differences in milk fatty acid profiles. Farm and management level differences in milk properties have also been observed elsewhere [81]. The exact reasons behind this are likely to be diverse and may include climatic conditions, genetics, and stress. However, perhaps the most notable and significant factor is nutrition. Doreau *et al.* [82] conducted a controlled study comparing the milk composition of mares fed on different diets and found that a variety of nutrients, but in particular fats and proteins, varied between groups. Deng et al. [83] also found dietary effects on milk composition, reporting that mares grazing on pasture produced milk with higher lipid concentrations. Indeed, variations in pasture nutrition will similarly impact the nutritional composition of milk.

Whilst there was no overall association of age with total SFA, MUFA or PUFA concentrations, it was significantly linked to one fatty acid, C18:0 (stearic acid). The negative relationship between C18:0 and age has also been observed in horses [84]. C18:0 is typically lower in horses than cow's milk, where a positive relationship with age has been observed [1, 85]. Notably, the mean concentration of C18:0 in thus study was 1.72%, which appears low compared to the 3.40% reported by Gregić *et al.* [86]. Barłowska *et al.* [42] reported C18:0 concentrations for Sokólski mares under different management and found mean concentrations to vary from 1.32% to 4.77%, putting the results presented here at the lower end of what may be expected.

DIM had strong and significant relationships with various fatty acids. Overall, SFA concentrations decreased and MUFA increased. The composition of milk typically changes over lactation to suit the changing nutritional demands of the growing offspring. Haddad et al. [87] reported reduced SFA concentrations and increased MUFA concentrations as lactation progressed. Similarly, Pietrzak-Fiećko *et al.* [88], studying three horse breeds, reported overall decreases in SFAs and increases in MUFAs from early, to mid, to late lactation.

Differences in SFA concentrations associated with DIM were primarily driven by C12:0 (lauric acid), C14:0 (myristic acid), and C16:0 (palmitic acid). C12:0 broadly contributes to the total SFA concentration of the milk but may also have anti-pathogenic properties [89, 90]. C14:0 can aid in fat accumulation, which is important for foal growth and fitness and similarly has been found to have anti-pathogenic properties [91]. C16:0 was the most abundant SFA with concentrations 4-5-fold greater than that of C12:0 and C14:0. The contribution of C16:0 to the overall fat content of milk is critical to foal growth and development, aiding in lipid assimilation [1]. The concentrations observed in this study were comparable to those observed in mare's wilk elsewhere [42]. Given the limitations of passive immunity in horses in utero, elevated levels of anti-pathogenic compounds early during lactation would be expected.

Regarding health indices associated with fatty acids, the mean AI value in our study was broadly similar to results reported elsewhere such as 0.83 from Mongolian horses [92], and 1.06 from Polish Coldbloods [40]. Alike, the mean TI value in this study was similar to those reported elsewhere, such as 0.37 reported by Chen et al. [92]. Both indices yielded values lower than those reported for cow's milk, which may range from 1.60 to 3.79 for AI and 0.39 to 5.04 for TI [93]. This suggests that mare's milk may pose less risk to cardiac health than cow's milk, which may be relevant to consumers at elevated cardiac risk.

### Immunoglobulin G concentration of CCH mare milk

The provision of IgG, via milk, from mares to foals is crucial for passive immunity. This is especially important in equines as the epitheliochorial structure of the placenta inhibits the transfer of immunoglobulins from mare to foal in utero [94]. This is compensated by high levels of immunoglobulins, notably IgG, in milk and especially in colostrum [95, 96]. Post-partum, colostrum IgG concentrations can undergo 20-fold concentration changes within 24h [97], highlighting how variable milk IgG levels may be. Milk IgG concentrations typically reduce as the foal matures towards weaning and as active immunity strengthens. Observed IgG concentrations in this study appeared to be broadly in line with those reported elsewhere for this stage of lactation, though given the focus on colostrum over milk and the high variability reported in the literature, there are limited opportunities to draw direct conclusions. In that regard, this study is one of few that report milk IgG levels this far into lactation.

Concentrations of IgG were greater in milk of mares from farm B than from farms A and C, though the reason for this is unclear. Dietary differences have been shown to influence horse milk IgG concentrations, in particular positive associations with dietary vitamin E [98, 99]. Only basic information was known about the diets and whilst both farms B and C provided pasture with salt-blocks, soil properties and the botanical composition of pastures may have been sufficiently different to yield differences in IgG. Elsewhere, maternal parity (number of foalings) has been associated with differing milk IgG concentrations [84], an effect not observed here.

## Conclusions

This study provides valuable insights into the nutritional composition and immunological properties of milk from the Chilean Corralero horse breed. It highlights the influence of various factors on milk composition and discusses its potential implications for foal health and human consumption. The study's sample size and diversity of origins make it one of the top-ranking studies in this field, ensuring a robust dataset for assessing the nutritional composition of CCH mare milk. Variations in milk composition, particularly in fat content and gross energy, were observed among farms. These variations were likely influenced by factors such as lactation stage and dietary differences. Despite differences in the number of days in milk between farms, which may be attributed to reproductive management, the origin of milk (breeding farm) independently affected milk composition. Factors such as the sex of the offspring and mare age also influenced milk components, with older mares tending to have higher milk fat content. CCH mare milk falls within the range of values reported for other horse breeds, but notable variations are observed in fat and total solids content. In comparison to human and cow milk, horse milk is richer in lactose and lower in fat and protein. The fatty acid profile of CCH mare milk also exhibits variations compared to other horse breeds, while the concentration of IgG varied among farms. Moreover, it is worth noting that mare milk poses less risk to cardiac health compared to cow's milk, as indicated by atherogenic and thrombogenic indices.

## Supporting information

**S1 Table. Pearson's correlation matrix.** Colored values are significant (p < 0.05) correlation coefficients (red: strong; yellow: medium; green: weak; in bold: negative). GE: gross energy; SFA: saturated fatty acids; MUFA: monounsaturated fatty acids; PUFA polyunsaturated fatty acids.
(TIF)

**S2 Table. Search in web of sciences and list of each reference studying mare milk composition.** N is number of mares sampled.
(TIF)

**S1 Dataset. Raw full dataset used for this study.**
(XLSX)

## Acknowledgments

We express our gratitude for the generous support provided by the staff of the institutional labs: the Laboratory of Immunology, Facultad de Medicina, for analyzing IgG, and the Animal Production Institute's Laboratory, Facultad de Ciencias Agrarias y Alimentarias, for analyzing milk energy, both from the Universidad Austral de Chile; the Laboratory of Feeds and Food, for analyzing forage and milk components, and the Laboratory of Fish Nutrition and Physiology, for analyzing fatty acids, both from the Departamento de Ciencias Agropecuarias y Acuícolas, Facultad de Recursos Naturales of the Universidad Católica de Temuco. The authors are also very grateful to the staff of the three farms for aiding in sample collection, to the farm owners for granting permission to conduct this study, and to the mares for allowing us to take the samples.

## Author Contributions

**Conceptualization:** M. Jordana Rivero, Roberto Leon.

**Data curation:** M. Jordana Rivero.

**Formal analysis:** M. Jordana Rivero, Andrew S. Cooke.

**Funding acquisition:** M. Jordana Rivero, Monica Gandarillas, Roberto Leon.

**Investigation:** M. Jordana Rivero, Roberto Leon.

**Methodology:** M. Jordana Rivero, Roberto Leon.

**Project administration:** M. Jordana Rivero.

**Resources:** M. Jordana Rivero, Monica Gandarillas, Roberto Leon, Alejandro Velásquez.

**Supervision:** M. Jordana Rivero.

**Visualization:** M. Jordana Rivero, Andrew S. Cooke.

**Writing – original draft:** M. Jordana Rivero, Andrew S. Cooke, Monica Gandarillas, Roberto Leon, Veronica M. Merino, Alejandro Velásquez.

**Writing – review & editing:** M. Jordana Rivero, Andrew S. Cooke, Monica Gandarillas, Alejandro Velásquez.

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
