## [Decision Letter · Decision Letter 0]

22 Jul 2024

PONE-D-24-22129Nutritional composition, fatty acids profile and immunoglobulin G concentrations of mare milk of the Chilean Corralero horse breedPLOS ONE

Dear Dr. Rivero,

Thank you for submitting your manuscript to PLOS ONE. After careful consideration, we feel that it has merit but does not fully meet PLOS ONE’s publication criteria as it currently stands. Therefore, we invite you to submit a revised version of the manuscript that addresses the points raised during the review process.

We look forward to receiving your revised manuscript.

Kind regards,

Mohamed Shaalan

Academic Editor

PLOS ONE

 [copy in funding statement].  

3. We note that Figure 1 in your submission contain copyrighted images. All PLOS content is published under the Creative Commons Attribution License (CC BY 4.0), which means that the manuscript, images, and Supporting Information files will be freely available online, and any third party is permitted to access, download, copy, distribute, and use these materials in any way, even commercially, with proper attribution. For more information, see our copyright guidelines: http://journals.plos.org/plosone/s/licenses-and-copyright.

4. We note that Figure 2 in your submission contain [map/satellite] images which may be copyrighted. All PLOS content is published under the Creative Commons Attribution License (CC BY 4.0), which means that the manuscript, images, and Supporting Information files will be freely available online, and any third party is permitted to access, download, copy, distribute, and use these materials in any way, even commercially, with proper attribution. For these reasons, we cannot publish previously copyrighted maps or satellite images created using proprietary data, such as Google software (Google Maps, Street View, and Earth). For more information, see our copyright guidelines: http://journals.plos.org/plosone/s/licenses-and-copyright.

Reviewers' comments:

Reviewer's Responses to Questions

**Comments to the Author**

1. Is the manuscript technically sound, and do the data support the conclusions?

Reviewer #1: No

Reviewer #2: Yes

Reviewer #3: Yes

2. Has the statistical analysis been performed appropriately and rigorously? 

Reviewer #1: No

Reviewer #2: Yes

Reviewer #3: Yes

3. Have the authors made all data underlying the findings in their manuscript fully available?

Reviewer #1: Yes

Reviewer #2: Yes

Reviewer #3: Yes

4. Is the manuscript presented in an intelligible fashion and written in standard English?

Reviewer #1: No

Reviewer #2: Yes

Reviewer #3: Yes

5. Review Comments to the Author

Reviewer #1: I'm sorry, but in my opinion the paper entitled Nutritional composition, fatty acids profile and immunoglobulin G concentrations of mare milk of the Chilean Corralero horse breed has serious problems in order to be accepted.

The paper has basic methodological problems that make it difficult to publish in its current form.

A work that aims to take into consideration the different factors that vary in milk quality (feeding, DIM, etc) should take into consideration sampling done in a more adequate way (sampling of the forage ingested and not the one available).

Furthermore, the statistical model adopted in my opinion is not adequate to evaluate the differences assessed.

Furthermore, there is no reference in the materials and methods how milk fat was quantified. Furthermore, the forage analysis lacks the evaluation of the ethereal extract.

Again, why do the authors report the evaluation of milk quality according to the sex of the foal? What is the influence of this factor?

Reviewer #2: Manuscript PONE-D-24-22129, entitled “Nutritional composition, fatty acids profile and immunoglobulin G concentrations of mare milk of the Chilean Corralero horse breed”

Recommendation: The above paper is not suitable for publication in its present form.

This review article provides information about the νutritional composition, fatty acids profile and immunoglobulin G concentrations of mare milk of the Chilean Corralero horse breed. It is in general appropriately organized, carried out and written, however there are some points that should be corrected or clarified.

L24: “The objective of the present study was to analyze…”

L33-34: “…mares' milk were within the reported range for other horse breeds but tended to have lower levels for fat and total…”

L36: Please delete “effect”

L43: “Milk of Mare (Equus caballus)…

L54: “…found to play a protective role against certain illnesses…”

L56: “Mare and human milk…”

L60: “help lowering”

L64-65: “Additionally, mare milk is characterized by the ability to protect…”

L76: “…and Shetland (17) in terms of milk solids…”

L88: What do you mean by “Rodeo symbolizes traditional cattle-working”?

L101: Please provide the number of approval

L105: Fig. 2?

L109: Please check figure and correct

L111: Fig. 2 or 3?

L119: Fig. 3?

L139-140: What do you mean by “a more regular air temperate”?

L149: “Samples were aseptically collected from the mares, at a volume of 250 mL per…”

L154: “collected” instead of “taken”

L163: “sampled” instead of “taken”

L164: “removed” instead of “taken”

L247: “among” instead of “between”

L249: These results are not shown in Table 2

Table 2: What about the Farm (B-C)?

L266: What do you mean by “(Error! Reference source not found.4)”?

L270: “…but greater lactose content (Fig. 4).”

L283: “affecting” instead of “driving”

L283: Please add the number of Table in the parenthesis

L285-286: Where are these data shown?

L290: The former or both?

L300: Inverse or negative?

L216: “Among these studies…”

L323-326: This sentence is not necessary. How were these percentages calculated?

L331: “Furthermore” instead of “More so”

L339: The duration of the gestation is a characteristic that is varied within breed and can be manipulated by the farmer?

L358-361: It is confounding as it is written

L370-371: “…suggesting that there are other factors than DIM imposing variation between the three examined breeding farms. This…”

L395: “…composition in horses.”

L412: “…how they vary with cow…”

L415: Please add a reference

L419: Please delete “of”

L425: “Lowe et al. (9) apart from Haflinger and Shetland Pony, and greater fat levels than that of Dutch Saddle…”

L432: “except for farm B”

L443: “observed” instead of “seen”

L448: In L444, you state that SFA prevail.

L452: “…horse milk has a low stearic…”

L454: “high” instead of “heigh”

L464: “…and 26.8 mg/g DM, respectively (77).”

L466: “…analyzed in our study contained these fatty acids.”

L479: “farm” instead of “fam”

L496: “…age has also been previously observed in horses (83).”

L517: “would make sense”?

L531: “sustain” instead of “see”

L549: “one of the top-ranking studies in this field”?

Reviewer #3: General opinion

In this study evaluated the various factors on nutritional composition and immunological properties of milk from the Chilean Corralero horse breed and discussed its potential implications for foal health and human consumption. The number of samples are very remarkable, this ensures the reliability of the results. The aim of this manuscript is very interesting for the readers; however, this manuscript must be improve, mainly the Results section.

Detailed opinion:

line 117: please add the amount of precipitation!

lines 130-131: the information in the brackets are better placed if put into the farm „A”!

Table 2 and 3: recommendation: all mean values of parameters would be shown in Table 2 and 3, not only overall mean and results of GLM!

line 250: MUFA is only found in Table 3!

line 266: mistake!

line 283: missing the number of a Table!

Figure 5: recommend inserting to the Figure a headline for better meaning!

6. PLOS authors have the option to publish the peer review history of their article (what does this mean?). If published, this will include your full peer review and any attached files.

Reviewer #1: No

Reviewer #2: No

Reviewer #3: No

---

## [Author Response · Author response to Decision Letter 0]

16 Aug 2024

A detailed note with the responses to the three reviewers have been uploaded as a Word document. The responses to each of the suggestions/questions are in blue text.

---

## [Decision Letter · Decision Letter 1]

5 Sep 2024

Nutritional composition, fatty acids profile and immunoglobulin G concentrations of mare milk of the Chilean Corralero horse breed

PONE-D-24-22129R1

Dear Dr. Jordana Rivero,

We’re pleased to inform you that your manuscript has been judged scientifically suitable for publication and will be formally accepted for publication once it meets all outstanding technical requirements.

Kind regards,

Mohamed Shaalan

Academic Editor

PLOS ONE

Reviewers' comments:

**Comments to the Author**

Reviewer #2: All comments have been addressed

Reviewer #3: All comments have been addressed

2. Is the manuscript technically sound, and do the data support the conclusions?

Reviewer #2: Yes

Reviewer #3: Yes

3. Has the statistical analysis been performed appropriately and rigorously? 

Reviewer #2: Yes

Reviewer #3: Yes

4. Have the authors made all data underlying the findings in their manuscript fully available?

Reviewer #2: Yes

Reviewer #3: Yes

5. Is the manuscript presented in an intelligible fashion and written in standard English?

Reviewer #2: Yes

Reviewer #3: Yes

6. Review Comments to the Author

Reviewer #2: Authors made the necessary amendments and I suggest the acceptance of their article. Some minor linguistic errors should be corrected

Reviewer #3: (No Response)

7. PLOS authors have the option to publish the peer review history of their article (what does this mean?). If published, this will include your full peer review and any attached files.

Reviewer #2: No

Reviewer #3: No

---

## [Editor Report · Acceptance letter]

10 Sep 2024

PONE-D-24-22129R1 

PLOS ONE

Dear Dr. Rivero, 

I'm pleased to inform you that your manuscript has been deemed suitable for publication in PLOS ONE. Congratulations! Your manuscript is now being handed over to our production team.

Kind regards, 

on behalf of

Dr. Mohamed Shaalan 

Academic Editor

PLOS ONE